# Emerging Perspectives of Blockchains in Food Supply Chain Traceability Based on Patent Analysis

**DOI:** 10.3390/foods12051036

**Published:** 2023-02-28

**Authors:** Jasna Mastilović, Dragan Kukolj, Žarko Kevrešan, Gordana Ostojić, Renata Kovač, Marina Đerić, Dragana Ubiparip Samek

**Affiliations:** 1Institute BioSense, University of Novi Sad, 21000 Novi Sad, Serbia; 2Institute for Artificial Intelligence Research and Development of Serbia, University of Novi Sad, 21000 Novi Sad, Serbia; 3Centre for Identification Technologies, Faculty of Technical Sciences, University of Novi Sad, 21000 Novi Sad, Serbia; 4Institute of Food Technology, University of Novi Sad, 21000 Novi Sad, Serbia

**Keywords:** food chain, traceability, blockchain application, database, patent portfolio, Latent Dirichlet Allocation, topic modeling, food safety, food authenticity

## Abstract

In the field of blockchain (BC) technology application in the food supply chain (FSC), a patent portfolio is collected, described, and analyzed using Latent Dirichlet Allocation (LDA) modeling, with the aim of obtaining insight into technology trends in this emerging and promising field. A patent portfolio consisting of 82 documents was extracted from patent databases using PatSnap software. The analysis of latent topics using LDA indicates that inventions related to the application of BCs in FSCs are patented in four key areas: (A) BC-supported tracing and tracking in FSCs; (B) devices and methods supporting application of BCs in FSCs; (C) combining BCs and other ICT technologies in FSC; and (D) BC-supported trading in FSCs. Patenting of BC technology applications in FSCs started during the second decade of the 21st century. Consequently, patent forward citation has been relatively low, while the family size confirms that application of BCs in FSCs is not yet widely accepted. A significant increase in the number of patent applications was registered after 2019, indicating that the number of potential users in FSCs is expected to grow over time. The largest numbers of patents originate from China, India, and the US.

## 1. Introduction

The food supply chain (FSC)is comprised of all the stages through which food products are delivered to consumers, from agricultural production, through storage, processing, distribution, and further toward food consumption, is a complex system which includes different operators, resources, and processes [1]. Economic performance, as well as the quality and safety of food products delivered to consumers through FSCs, depends on inputs from all FSC stakeholders: farmers, warehouses, processors, logistics, distributors, wholesalers, and many others [2]. With the development of more complex FSCs, tracing and tracking of food from raw materials to the very products delivered to the consumer, with disruptions, fragmentation, poor product traceability, improper product flows, food contamination, and food recall, have become the prevailing issue of FSCs, demanding much effort from all stakeholders, and coming into the focus of both consumers and policymakers [3]. Common to all supply chains is that they could benefit from implementing traceability [4], especially in reverse supply chains when unused, faulty, or damaged products are returned to the manufacturer in an attempt to improve quality [5]. The weakness of all supply chains, including supply chains of pharmaceutical products, courier express, luxury goods, consumer electronics, manufactured goods, automobiles, textile, wood, dangerous goods, and particularly agri-food products, is that transparent information is not currently offered to the final consumer as part of supply chain management [4].

As a technological concept that evolved from the first cryptocurrency, blockchains (BCs) are increasingly entering into many areas of economics [6]. A BC is a database consisting of encrypted blocks of records in a distributed public ledger of digital events shared among participants of a chain which enables future verification of all executed transactions. It is globally used as a robust tool for supporting financial systems, but its use in other fields of economics is being widely considered, particularly regarding tracking in the supply chains, resulting in its expansion [7,8,9]. BC technology facilitates security, privacy [10,11] transparency, decentralization, and immutability [12], thus enabling trustful, secure, and authenticated systems of logistics and information exchange to be established in supply networks [6].

Insight into the capabilities of blockchain-based traceability solutions come from the academic literature, in which it is pointed out that both tracking and tracing via BC applications contribute to better transparency in supply chains and are thus gaining in popularity, particularly in food and pharmaceutical supply chains. The conclusion, however, is that, in spite of numerous applications reported in the literature, most of them are conceptual in nature, and real implementations of BC-based traceability solutions are very rare [4]. Thus, during the past few years, the application of BC technology in FSCs has come into the focus of numerous researchers. BC technology is proposed as a solution for tracing in FSCs, regarding their safety, immutability, transparency, and scalability [13], with the ability to follow a product along its lifecycle [14], including internal traceability in food processing [15] and enhanced mechanisms for control and response to food recalls in the FSC [16]. Wang et al. [17] emphasize that application of BC technology contributes to FSC digitalization and disintermediation with extended visibility, traceability, and improved data security, while Balamurugan et al. [18] state that blockchain technology may support the identification of unsafe or fake foodstuffs, in order to block further access to them. When proposing a framework for traceability within FSCs and addressing the key challenges for FSC stakeholders, Baralla et al. [1] emphasized BCs, in combination with IoT, as technologies of choice for the development of such a system. An easy and simple methodology for integration of BC technology in FSCs that allows traceability and provides the consumers with sufficient information to make informed purchasing decisions was proposed by Bettín-Díaz et al. [19] based on best practices engineering, technology, and marketing.

BC technology has also been proposed as a solution for the protection of an increasing number of IoT-oriented applications [20], product-selling platforms, on-demand service platforms [21], and cloud computing solutions [22] developed for, or applied in, the FSC. Advantages brought into FSCs through the introduction of BC technology are widely discussed in academic literature; ensuring the integrity and privacy of datasets even when they are released to the public [20], counterfeit of hardware and software faults, security issues during communication, system management difficulties [23], internet stability and security complications [24], data security [23] accountability, protection, neutrality, and efficiency of processes [24] are just a few examples. However, the fact that diverse barriers, preventing the expansion of BC technology application in FSCs [25], still exist is widely elaborated in the academic literature [12,17,26,27,28].

Patents can be used as a detailed source of both quantitative and qualitative technological information [29] and are thus widely used for monitoring technology trends [30] and industry competitiveness [31]. Reviews based on the analysis of patent portfolios have been published in different technological fields, such as, for example, the application of carbon nanotubes [32], battery technologies for electric mobility [33], telecommunications [34], artificial intelligence in the automotive industry [35,36], multi-compartment refrigerators [37], new space missions [30], and other emerging [38] or promising [39] technologies. Among numerous methodologies for the exploration of technology trends based on patent data and topic modeling, a method based on the use of hierarchical probabilistic models for uncovering patterns of words has been proven to be among the most effective in discovering and classifying hidden topics behind patent documentation [34,40].

In the present study, we sought to gain insight into the technology trends in the field of BC technology application in FSCs via collection of patent portfolio, which was analyzed to determine application trends and distributions and then subjected to topic modeling to identify latent topics and relate them to development trends and opportunities in FSCs.

## 2. Materials and Methods

### Review Approach

Extraction of the patent portfolio from patent databases was performed using the commercial software platform PatSnap [41,42]. Both active granted patents and pending patent applications were extracted. Patent title, abstract, and body text were searched using the following terms: ((blockchain OR “block chain” OR blockchains OR “block chains”) AND (traceability) AND (“food supply chain” OR “food value chain” OR “agri-food chain” OR “food supply chains”)). Extracted patent attributes included application number, title, classification code, priority and application date, abstract, family count, legal status, and the number of backward and forward citations (Figure 1).

A patent portfolio of 82 documents was created, among which 26% were active patents, 70% were pending patent applications in different phases, and 4% of documents were undetermined. The patent application trends and the application authority structure was analyzed for the created patent portfolio.

For the interpretation of the obtained unstructured document collections, topic modelling based on Latent Dirichlet Allocation (LDA) was performed [43]. The documents were parsed prior to data processing. The text in abstracts were pre-processed in the following manner:-Removal of punctuations, digits, words, and tags shorter than four characters-Conversion of letters to lowercase-Filtering of words not contributing to the meaning of the phrase (stop words)-Reduction of words to their roots using the Porter stemming algorithm [44]

The pre-processed abstract texts were used to group patents in the portfolio into technology topics using Latent Dirichlet Allocation modeling (LDA) [45]. In LDA, documents are viewed as random mixtures of different topics, where each topic is characterized according to the distribution of words. The topic distributions in documents share the common Dirichlet prior α, and the word distributions of topics share the common Dirichlet prior η. Given the parameters α and η for document d, parameter θ_d_ of a multinomial distribution over K topics is constructed from Dirichlet distribution θ_d|α_. Similarly, for topic k, parameter β_k_ of a multinomial distribution over all words is derived from Dirichlet distribution β_k|_η. The generative process of the LDA method is given in Table 1.

The number of topics embodied in the patent dataset were selected based on evaluation of the predictive likelihood of models with a different number of topics and constructing the predictive likelihood curve (Figure 2). Bearing in mind the rather small number of patent documents in the portfolio, the first point where the curve changed direction (4 topics) was selected as the optimal number of topics.

Based on the LDA distribution results of documents from patent portfolios across four latent topics (i.e., technology fields), the probabilities of participation of each topic in the considered documents were obtained as values ranging from 0 to 1 (Figure 3). Each document represents a probabilistic mixture of topics generated from the patent collection contents, and the patents in the portfolio were each assigned the topic with the highest calculated probability.

Principal component analysis was used to visualize the distribution of patents from the portfolio across LDA topics.

As LDA does not provide topic labels, identified latent topics were initially labeled as topics A, B, C, and D. To reveal the logical content and name, the identified latent topics content of patents in each topic were manually checked, the most frequent terms associated with each topic by LDA were taken into consideration, the most frequent patent classification codes were analyzed, and we proposed titles that best described each topic. We manually classified the most frequent words, identified by LDA as nouns, verbs, adverbs, and other words, to enable understanding of the common features for each topic. The International Patent Classification (IPC) system was used, given it is the most frequently used system to hierarchically organize the structure of technical fields and technological areas of a patent. The patent, depending on the technical broadness of the covered invention, may have several IPC codes assigned to it, and all assigned IPC codes were taken into consideration in the analysis.

The outcome of the LDA was used to draw a dynamic picture of the development of BC technologies application in FSCs, with analysis of geographical distribution, and identification of the most valuable patent applications was based on patent forward citations and patent family size [32,46,47,48].

In the cited works, influential patents having dominant technology or high applicability are distinguished and, accordingly, were used as an indicator of a patent’s technological significance. A simple patent family representing a set of patents applied in various countries to protect a single invention, besides representing the importance of technology, also reveals the width of its potential market [32,46]. A larger patent family is indicative of the attractiveness of the protected invention or that the technology is more innovative. Obtained results were used as the base for identifying the potential drivers and paths in future development of BC applications in FSCs.

## 3. Results

### 3.1. Patent Application Trends

The annual count of published patent applications of a patent portfolio in the field of BC application in FSCs (Figure 4A) shows that the first patents in this field were filled after 2015, while significant growth of patent applications started from 2019, and the number of applications increased exponentially from this year onwards. Considering that application of the BC technology actually began at the end of the first decade of this century, such a trend is expected and understandable. It can be concluded that the growth in the practical application of BCs in FSCs is still in the initial phase, with significant growth expected in upcoming years. An increase in the application of BCs in patents related to FSCs from 2019 is in line with the results of Nan et al. [49], whose results indicate a significant increase in BC-based patent applications in the US and China since 2017.

In the geographic distribution of application authorities of the observed patent portfolio (Figure 4B), it is quite interesting that the leading countries with the highest shares of patents are China (38% of the total number of patent applications) and India (28% of the total number of patent applications), while the United States, with 20% of the total number of patent applications, is only in third place. The remaining 21% of patents are registered elsewhere in the world, with the European Patent Office and Australia taking both 7% of initially filed applications and the rest of the world claiming the remaining, only 7%.

### 3.2. Modeling of Portfolio Technology Space Based on the LDA

Based on the LDA, patents from the portfolio were found to be distributed across four latent topics. The first step in the analysis of obtained results was the visualization of relations among formed topic clusters in a scatter plot presented in the factorial plane using PCA (Figure 5). It is obvious that patents from the analyzed patent portfolio are distributed in four mutually distinguished topics. Topic A, placed in the lower right quadrant of the factorial plane, is distinguished from the other three topics in relation to being the first, most influential dimension of principal component analysis, explaining 42.21% of the total variability. Topic C, positioned in the lower left quadrant, is quite similar to topic A, in terms of being the second PCA dimension, while topic D, and particularly topic B, are differentiated from topics A and C in that this PCA dimension explains 35.83% of the total variability.

Analysis of the content of identified latent topics was performed based on analysis of the most frequent words revealed from the LDA, and the most frequent IPC codes attributed to the patents.

Frequencies of occurrence of the most frequent words in abstracts of patent applications, allocated to four topics (Figure 6), provide a step forward in understanding the underlying trends in issues of patenting activities in which hidden topics occur. The most frequent words revealed for each topic from LDA were further classified into three groups of words and are presented in Figure 6 with different colors. The first group of words are verbs, which most probably indicate which operation, action, or activity in FSCs are being addressed in the patents (colored in green). The second group is nouns, which most probably indicate the subject or object in FSCs targeted by the patent (colored in dark gray). The third group of words are revealed as the most frequent and most common for all topics, i.e., the words from the patent database search pattern, or the adverbs that have not been taken into consideration in the analysis of issues targeted within topic clusters of the patent portfolio. Verbs and nouns identified as the most frequent in each topic are further systematized in Table 2, in which data used for revelation of hidden topics are presented.

The second step forward toward understanding the underlying trends regarding the occurrence of issues that are latent topics of patenting activities, was the analysis of the most frequently assigned IPC codes within topics (Table 1). Table 1 also contains explanations of the most frequent IPC subgroups in the portfolio, together with the frequency of their appearance in the patents assigned to each hidden topic. However, within all topics, the most frequently assigned IPC codes were, as expected based on patent portfolio search criteria, with the IPC group G06Q, described as “Data processing systems or methods, specially adapted for administrative, commercial, financial, managerial, supervisory or forecasting purposes; systems or methods specially adapted for administrative, commercial, financial, managerial, supervisory or forecasting purposes, not otherwise provided for”.

Further analysis was directed toward understanding the focus of each of the four identified hidden topics in the analyzed patent portfolio. Besides data from Table 2, including the most common words in the abstracts and the most frequently assigned IPC codes, the analysis also included an in-depth review of the patents assigned to each topic by LDA, which we performed manually.

Topic A: The IPC code most frequently assigned to patents on topic A is G06Q10 (Data processing systems: administration, management), followed by G06Q30 (Data processing systems: commerce), and G06Q50 (Systems or methods specially adapted for a specific business sector) (Table 2). This indicates that classified patents, as being on topic A, are related to data processing systems that support administration, management, and commerce in a specific business sector, which, in this study, is the food sector.

The structure of the words most frequently occurring in topic A (Table 2) points out that the patents assigned to topic A are related to tracing and tracking of quality and safety of food products and production processes on their way to consumers.

Patents on topic A are mainly oriented to the technical application of blockchain technology (CN114511333A, CN114722117A, US20220198446A1), in combination with the different labeling solutions of QR code (CN105868995), two-dimensional labels (CN108681843A, US20220230134A1), dot-matrix labels (US20200279365A1), DNA tags (US20190285602A1), RFID (CN208400175U, US20220230134A1), NFC (CN105868995A), and different configurations (EP2374017B1). The technical part of topic A includes patents describing the collection of data (IN202011040078A, EP3891501A4).

Patents within topic A also include blockchain applications for food safety data collection and tracing (CN114722117A) in different industry sectors, such as the grain and oil sectors (CN114386992A) or the meat industry, with the application of networks of sensors for data collection (CN105868995A), including monitoring from stable to table (CN208077495U), and structuring of data according to HACCP (CN109657996A) principles. Some patents involve decision-making modules that, based on input data, can estimate “pollution score” (US11282023B2) or even recall food products from the market (US20220215353A1).

Some patents on topic A are dedicated to the provision of information to consumers, including information about products collected in the food chain (WO2022140465A1, US20210366586A1) or information about prepared meals (EP3899831A1).

Based on the provided information, Topic A is labeled as BC-supported tracing and tracking in FSCs.

Topic B: The IPC codes most frequently assigned to patents from on Topic B (Table 2) are G06Q10 (Data processing systems: administration, management), G06F16 (Digital computing or data processing equipment or methods, specially adapted for specific applications), and G06Q50 (Systems or methods specially adapted for a specific business sector). Based on this information, it is obvious that, for topic B, the patents related to data processing systems supporting administration and management in FSCs with digital data processing equipment, systems, or methods are included.

The structure of words most frequently used in the patent abstract, in the case of patents assigned topic B, indicates that, in these patents, the issues of connecting, detecting, improving, predicting, or supporting of operations in FSCs are treated on the level of the enterprise, storage, cabinet, meal, or cell (Table 2). Based on this information, it seems that patents assigned topic B are most probably related to novel equipment, methods, or systems through which detection, prediction, improvement, or other operations at different levels in FSCs is achieved.

Revision of patents on topic B reveals that these patents focus on the description of new apparatus, devices, or methods that can be used for different purposes in different phases of a FSC. New devices and methods described in topic B are: pathogen sampling device (CN114190325A), conversion conveying device (CN216970928U), quality safety sampling and inspection device (CN113552297A), temperature and humidity detection alarm device (CN110542452A), food supply chain hazard prediction method and device (CN113378383A), and improvement of Byzantine consensus method based on food supply chain (CN112330238A). Some patents also deal with new devices for food distribution (CN206451226U, CN103956001B). 

Based on the provided information, topic B is labeled as devices and methods supporting BC application in FSCs.

Topic C: The IPC codes most frequently assigned to patents on topic C are G06Q10 (Data processing systems: administration, management), G06Q50 (Systems or methods specially adapted for a specific business sector), and G06Q30 (Commerce, e.g., shopping or e-commerce) (Table 2). In patents on topic C, the issue of data processing systems and methods in FSCs is upgraded with commercial aspects.

The structure of the words most frequently used in patent abstracts, in the case of patents assigned to topic C, indicates that, in these patents, issues of the utilization of novel technologies, such as artificial intelligence, IoT, sensor networks, and other digital solutions for securing, managing, accessing, or learning in agriculture on the level of a farmer, farm, or storage are the focus (Table 2). This information points out that patents assigned to topic C are most probably related to a combination of BCs with other innovative ICT solutions, such as artificial intelligence, sensor networks, Internet of Things, and others.

Indeed, analysis of the patent portfolio for topic C reveals that, besides the application of BCs in FSCs, common to patents within this topic, is the use of other novel technologies to improve the use of blockchains in food applications, including IoT (AU2021107272A4, IN201911048905A, IN202121045069A, AU2021106931A4) or a combination of artificial intelligence with IoT (AU2021103789A4, IN202241021982A) for the achievement of diverse management goals, such as, for example, to develop new apparatuses for detecting quality and shelf life of processed seafood (IN201941005285A) or Smart Public Food Distribution Management Systems (IN202241000876A).

Based on the provided information, Topic C is labeled as combining BC and other ICT technologies in FSCs.

Topic D: The IPC codes most frequently assigned to patents on topic D are G06Q20 (Payment architectures, schemes, or protocols), G06Q30 (Commerce, e.g., shopping or e-commerce) and G06Q10 (Data processing systems: administration, management) (Table 2). According to such assignment of IPC codes, patents on topic D focus more on blockchain application in supporting commercial aspects of FSCs, payment protocols, and e-commerce.

The structure of the words most frequently used in the patent abstract, in the case of patents assigned to topic D, confirms this statement, as they indicate that, in these patents, the issues of e-commerce and online shopping and purchasing, as well as the issues of transactions, delivery, packaging, and distribution processes for fresh commodity and other units are the focus (Table 2).

Analysis of patents on topic D confirms that they include distribution of food, including intelligent shopping baskets (CN113077314A), purchases incented by donations (US17/323427), end-to-end food delivery systems (US17/249142), platforms for distribution and shopping on-line (CN202111208284.X), and fresh food e-commerce supply chains (CN202111336297.5).

Based on the provided information, topic D is labeled as BC-supported trading in FSCs.

### 3.3. The Most Valuable Patents or Applications within Topics

For the studied patent portfolio, forward citations and patent family sums, as well as the sum of these indicators, were analyzed (Figure 7). Because patenting of BC-based inventions intended for FSCs started relatively recently (Figure 4), the value indicators for most patents is consequently quite low. In the case of forward citation, such a result is expected, as there has not been sufficient time for this indicator to grow. On the other hand, low values of family size confirm that patents related to the application of blockchains in FSCs do not yet address the wider market.

In order to gain insight into the potentially most influential patents from the portfolio, and particularly into their distribution among topics revealed from LDA, 16 patents with the highest scores of observed value indicators are presented (Table 3) by identified topics.

The most outstanding patent regarding value indicators (CA2776577) is on topic C, to which patents combining blockchain and other ICT technologies in FSCs are assigned. This patent presents a system and method for establishing an agricultural pedigree for at least one agricultural product.

The highest number of patents with relatively high-value indicators are on topic A, to which patents related to BC-supported tracing and tracking in FSCs are assigned. Such an outcome is quite understandable, since traceability in the FSC is required in food-related regulations worldwide.

### 3.4. Distribution of Topics by Country of Origin

Leading countries holding patents denoted as being topic A, which deals with BC-supported tracing and tracking in FSCs, are the US and China, while a significantly smaller number of patents on this topic have been filed in India, the EU, and other countries (Figure 8).

Regarding topic B, China represented 77% of patents, which implies that this country takes the leading role in the development of novel devices and methods supporting the application of BCs in FSCs (Figure 8).

The highest number of patents on topic C, related to combining BC and other ICT technologies in FSCs, was submitted in India with 71% of the total number of patent applications. A significant number of patent applications from this topic originate in Australia (18%), while only 11% come from the rest of the world (Figure 8).

The highest share of patents on topic D, which deals with BC-supported online trading in FSCs, comes from China (45%), closely followed by the US (33%) (Figure 8).

## 4. Discussion

### 4.1. Patenting Trends and Patenting Landscape

The first fact confirmed in the present investigation is that patenting activities in the field of application of BCs in FSCs were initiated quite recently, and the number of granted patents is very low. The number of applications started to increase during the last four years, while the value of the patents, in terms of forward citations, is quite low for the majority of patent applications. An exponential increasing trend in BC-related patenting in general started in 2017 and reached 14072 patents in 2021, from 231 in 2016 [49]. Patent analysis in other areas shows similar trends, but with differences, in terms of patents targeting different fields of application. For example, BCs on 5G patent portfolios consisted of over 1000 public and over 120 approved patents in the periods 2014 to 2020 [50], while, for BC and energy, a study from 2020 revealed 24 approved and 295 public patents [51].

Additionally, the low number of patents with high value in terms of family size points out the absence in global spread of the developed solutions. These observations are aligned with findings from the academic literature, stating that, despite the enormous potential for large-scale improvements in FSCs through the utilization of BCs [6], its adoption in FSCs is still in the introductory phase [52], and successful projects and sector-specific studies of BCs in FSCs are still scarce [53]. The situation of BC technology application in other supply chains might be a good indicator of probable development in FSCs. Patent life cycle analysis results indicate that patents related to BCs based on 5G peaked in 2018, followed by a bottleneck period beginning in 2019 [50]. Patenting in the cluster of smart contract applications, privacy preserving and intellectual property, certificate issuance and verification, tokenization, virtual reality gaming, and interoperability is also in a stage of maturity [49]. Such findings indicate that similar fast development might also be expected in the case of BC application in FSCs.

However, governance and broader partnership issues required for successful, sustainable applications have still not been solved [46], and BC technology, although demonstrated as a good solution in some FSCs, is not yet ready for mass acceptance. This situation is the consequence of different barriers associated with the implementation of BC technology in FSCs, which are still present in practice. These barriers include technological, organizational human resources-related, and economic barriers [26]. The barriers and challenges are diverse and related to human capital, such as gaps in understanding [17], lack of skills and training [54], and an insufficient number of trained members in the workforce [27]. Barriers are related also to technical issues in application [55], including scalability [53,54], and development and adoption of the most appropriate BC structures for each FSC [13]. Economic barriers include a lack of opportunities for further research [17], availability of funds and clarity on economic gains [27], and incentivization [53], as well as policy barriers, such as lack of regulations [54]. Worth mentioning also are those barriers related to the interest of stakeholders in preserving established hierarchies [6] and their unpreparedness to accept increased obligations related to fraud prevention, as well as waste, losses, and environmental impact reduction [25].

Another interesting observation is related to the countries that have been depicted as leaders in patent activities in the field of BC application in FSCs. Namely, the present analysis highlighted India and China as the countries with the highest number of patents dealing with the above-mentioned issue. The identified situation regarding geographical distribution of patents dealing with BC in FSCs is somewhat different from that of the general distribution of patents related to BCs. In 2021, the largest share of BC-related patents originated from China (36%), followed by the US (26%), while India represented only 2% of patents [49]. When it comes to China, one of the possible explanations for the expansion in this area, and quite possibly in related areas, might be an adoption of the country’s strategic development, which will ensure its independence in the IT domain from foreign technologies [56]. Regarding India, the patents are primarily focused on applying BCs in combination with other internet technologies, which can be attributed to the rapid development of the IT sector in that country and its orientation toward export [57].

### 4.2. Application Areas

It might be interesting to analyze the drivers that resulted in the development of paths, represented by LDA-identified topics, in development and patenting in the field of application of BCs in FSCs.

The most frequently assigned IPC to the patents related to BC application in FSCs was G06Q, for all identified latent topics. However, the analysis of 335 patents related to BC technology in general awarded in the US, as of 2020 [58], revealed that the most frequents IPC codes were H04L (transmission of digital information, e.g., telegraphic communication) present in 84% of patents, followed by G06F (electric digital data processing) present in 53% of patents, while G06Q (data processing systems or methods, specially adapted for administrative, commercial, financial, managerial, supervisory, or forecasting purposes) was assigned to 33% of analyzed patents. This observation points out the somewhat specific focus of patents related to BCs in FSCs.

BC technology is characterized by quite wide possibilities for application in different fields. Backman et al. [49] identified that, beside financial transactions in cryptocurrencies, integrity verification in insurance, data management in human resources, privacy and security in secure storage, reputation of education, governance of e-voting, integrity of IoT business, electronic health records, and, finally, business industry supply chain are also prospective fields for BC application. For BCs in FSCs as the subfield, the latest research by Zhao et al. [59] points out, based on findings from the academic literature, that applications have been developed in four main directions comprising, in addition to traceability, manufacturing, water management, and information security.

In the present research, we confirm that the most developed field of BC application in FSCs is related to traceability enhancement. Considering that traceability represents a legal obligation in FSCs, it is quite understandable that the largest number of patent applications is registered in the domain of application of BC technology for tracing and tracking along FCSs [60]. Achieving internal traceability within a company involved in FSCs requires extensive engagement of human resources, as well as continuous record-keeping. Those processes can be significantly facilitated by applying contemporary digital technologies [61]. In terms of external traceability along the FSC, the issue of trust and confidence in records is circulating between companies in the FSC [62]. Thus, BC technology represents a logical solution, since it is designed to overcome such problems. It also reduces the level of human engagement while increasing the level of confidence, not only among FSC members, but also among consumers.

Another important element that additionally simplifies the process of data collection within FSCs are devices and instruments that enable direct upload of measured or registered data into digital databases (EP3520350B1). The application of such equipment and its coupling with BCs, as a technology-enabling reliable collection and storage of original data, represents a prerequisite for further progress in this area. These circumstances have logically contributed to the development of technical solutions and their patenting in the field of development of devices and methods which support the application of BCs in FSCs.

There are numerous platforms, IoT applications, patents using artificial intelligence, and sensor networks that have been developed to enable communication and facilitate specific operations and transactions in FSCs [53,63]. These systems use or generate a significant quantity of important information. Confidence in and reliability of these data emerge as an issue, representing in many cases an obstacle for their acceptance and application on a large scale [63]. In this case also, BCs represent an ideal solution and, thus, a driving force for patentable innovation in this domain.

The authors of [64] also agree that it is important that the introduction of blockchain technology in FSCs is not limited to this single technology, but rather to a whole stack of new technologies, including sensing technologies, big data analytics, artificial intelligence, Internet of Things robotics, digital twins, cyber-physical systems, 3D printing, and others. As the patents on topic C are mainly dealing with the implementation of IoT and other novel technologies, it could be assumed that development IT (development of mobile networks and accessibility of satellite communication) can be used to overcome remote agricultural production, but also remote consumption in Australia, while development of IT communications in India allows for overcoming the problem of missing conventional infrastructure.

Finally, online trade is becoming part of the daily activities in many economic sectors, including food [63], and the application of BCs as a technology supporting online trade solutions is, therefore, another tenable field of innovation and patenting.

Renda [64] agrees that benefits of introducing BC technology in FSCs may address much wider issues related to food security and safety and human health in general, reduction of food waste and losses, animal welfare and climate change [65], as well as the development and digitalization of rural areas [66].

### 4.3. Further Prospects

The structure of the analyzed patent portfolio reflects realistic needs for improvements in FSCs, as well as the developmental needs related to the enabling of BC applications to address these needs. The increasing trend in the number of patents related to the application of BCs in FSCs during the recent few years indicates the possibility for increased application of BCs in FSCs. The progress of BC applications in FSCs in the future also depends on the implementation of measures aimed to remove identified barriers for the application of BCs in FSCs. Regulations introducing BCs as a technology to support traceability in FSCs, as well as regulatory arrangements related to this issue, would undoubtedly support the process of introduction of BCs in FSCs.

Another direction toward widening of the application of BCs in FSC might be the identification of stakeholders in FSCs motivated to voluntarily adopt the implementation of BCs as a tool for improvement of their market position. Some of the early potential adopters of BC technology in FSCs are depicted in the academic literature, and they can be divided into two groups.

The first group represents the producers of added-value food, motivated to improve the traceability of their products by using BCs, since it will enable them to more easily secure confidence in origin or other quality trade which separates their product from other similar products and provides all the necessary data to prove it more easily. This group of producers will benefit from consumers’ confidence in the origin and quality of their products. Therefore, this group of users has a relatively high probability of spontaneous implementation of BC solutions. Representative users from this group are producers of organic products [55,67], halal food [68,69], and traditional food [61]. This observation is supported, also, through the fact that almost half of the initiatives and projects registered in the field of BC application in FSCs [70] is targeting food integrity.

On the other hand, the main goal for the application of BCs in the second group is to secure food safety and traceability in complex FSCs. Some stakeholders will implement BC technology to prevent or detect fraud and trust issues, as in the case of pork and mango in supermarkets [71], while others will improve traceability to further improve quality. Examples of the latter are dairy production [72,73,74], edible oil production and distribution [75], and beef production [76].

The fact that BC technology is attracting the attention of numerous stakeholders is also confirmed through the existence of many projects and initiatives that aim to contribute to the establishment of a trusted environment, are the building of more sustainable, more transparent, and more integrated FSCs. Based on the analysis of projects and initiatives related to BCs in FSCs targeting FSC issues, besides the dominant issue of food integrity (50% of identified initiatives), there were also issues of supervision and management of FSCs (14.5% of identified initiatives), waste reduction and environmental awareness (10%), support of small farmers (16%), food safety (6%), and food security (4%) [71].

To strengthen the further application of BCs in FSCs, it is necessary to clearly emphasize the contribution of this technology to the economic performance of FSC stakeholders and FSCs as a whole. Additional funding of research targeting technical barriers and obstacles, as well as the development of educational and promotional strategies, might also contribute to further growth of BC application in FSCs. Finally, the number of potential users in FSCs is expected to grow over time, as the development of BC technology provides numerous benefits that will become more significant than the mere mandatory effect of its application. A most recent study providing content-based analysis of the academic literature [77] revealed that BC application in the agrifood sector is a relatively recent topic that started to gain traction in 2018, when the research community started intensively dealing with this issue, with more than 75% of identified related records in the academic literature being published in 2020 and 2021. The identified sharp increase in the number of patents and patent applications related to BC application in FSCs is a reflection of vigorous research activities, resulting in ready-to-implement solutions in this field. The obtained results will support research community members in making decisions related to the development of high-readiness level products in the field of BC application in FSCs.

The main limitation related to present research is the lag period of patents, which has certainly contributed to incomplete results within the past year. The second limitation is related to the fact that patent application does not necessarily result in an approved patent, which was taken into consideration, but, due to the low number of approved patents, they had to be taken into account for this analysis. However, the clearly identified increasing trend in the number of patents suggests that these limitations most probably do not affect the obtained results to a great extent.

However, the development of BC applications in FSCs is closely related to integration with other information and communication technologies, in particular IoT. Finding that combination of BC technology with other information and communication technology, and IoT in particular, has been adopted for the improvement of FSCs was already emphasized in [59].

## 5. Conclusions

BCs have great potential for application in FSCs. Numerous research activities have focused on the possibility of applying BCs in FSCs, but, as a consequence of numerous barriers, this technology is not yet widely applied in FSCs. The process of patenting in the field of application of BC technologies in FSCs started relatively recently, and the highest number of patents originate from China, India, and the US.

The analysis of latent topics using LDA indicates that inventions related to the application of BCs in FSCs are patented in four key areas: (A) BC-supported tracing and tracking in FSCs; (B) devices and methods supporting the application of BCs in FSCs; (C) combining BCs and other ICT technologies in FSCs; and (D) BC-supported trading in FSCs. The countries of the patent applications differ significantly by topic, with China leading, in the case of BC-supported tracing and tracking (A), development of devices and methods supporting the application of BCs in FSCs (B), and BC-supported trading in FSCs (D), while, in the case of combining BCs and other ICT technologies in FSCs (C), the dominant share of patents is from India. In the case of the US, patent activities are significant in the case of BC-supported tracing and tracking (A) and BC-supported trading in FSCs (D).

Based on the indicators, the value of patents were relatively low, which, in the case of patent forward citation, can be attributed to the short time that has elapsed since patent application, while, in the case of family size, it confirms that the application of blockchains in FSCs has not yet been widely accepted, and it is still rather a matter of solving case-level challenges.

The most influential are patents on the topic area related to BC-supported tracing and tracking in FSCs, which can be attributed to the fact that traceability in FSCs is a legal obligation for all entities involved in food supply, which further increases the need to simplify the extensive work of food tracing and tracking and make it more reliable by applying new IT solutions.

The exponential growth in the number of patent applications in the field of BC technology application in FSCs suggest that a significant increase in the adoption of this technology can be expected in the near future. Development and implementation of measures for the mitigation of barriers related to the introduction of BC technology in FSCs would undoubtedly accelerate this process.

## Figures and Tables

**Figure 1 foods-12-01036-f001:**
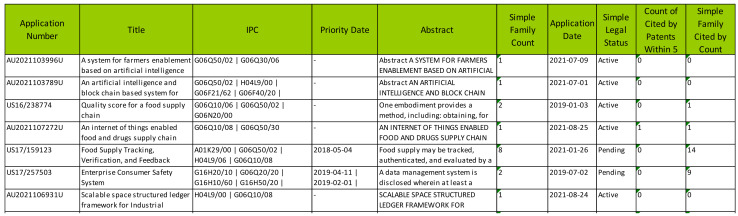
Screenshot of an example of a relational table for collected patent documents.

**Figure 2 foods-12-01036-f002:**
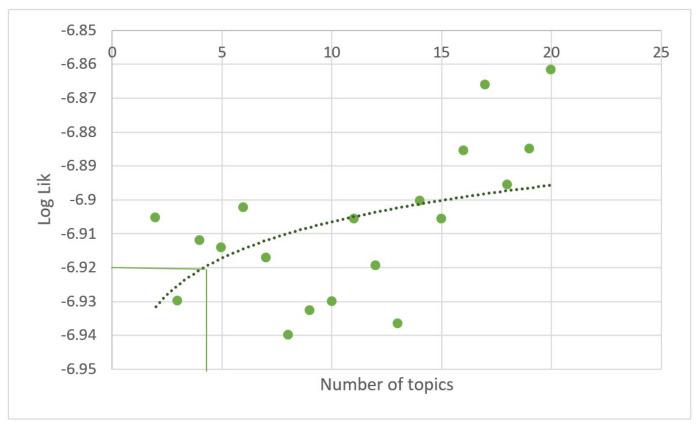
Predictive likelihood of models with different number of topics.

**Figure 3 foods-12-01036-f003:**
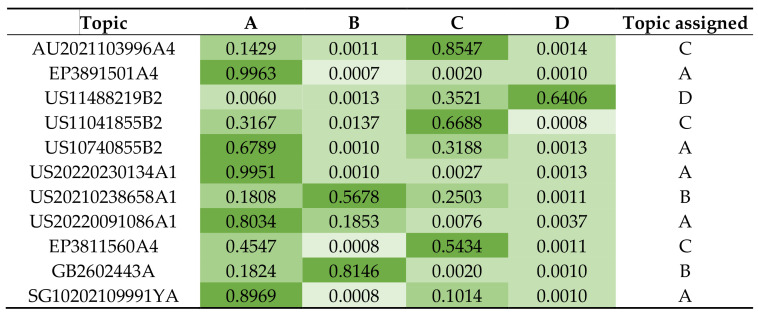
Example of topics heat map resulting from the LDA Analysis.

**Figure 4 foods-12-01036-f004:**
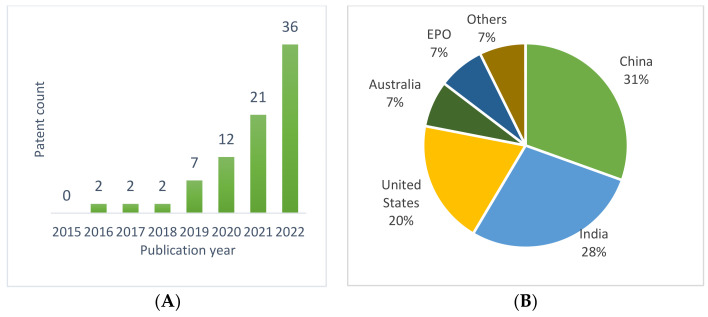
Patent portfolio publication trend (**A**) and authorities of the portfolio applications (**B**).

**Figure 5 foods-12-01036-f005:**
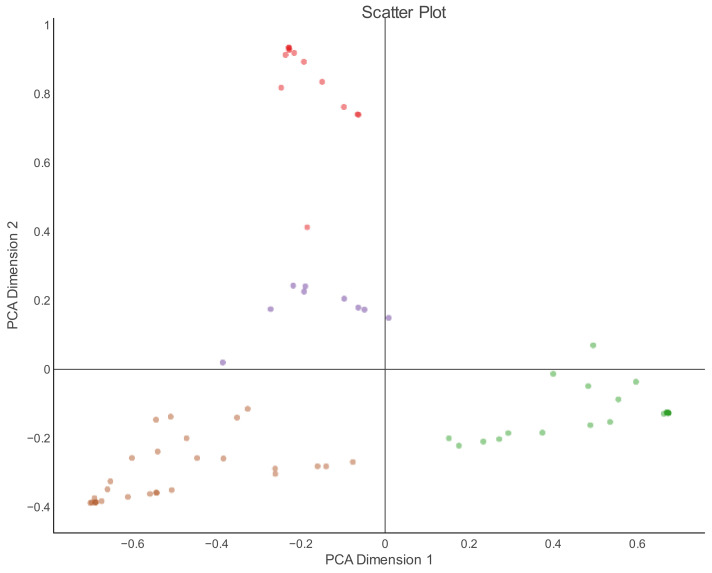
PCA-based distribution of patents in four LDA topics in the factorial plane.

**Figure 6 foods-12-01036-f006:**
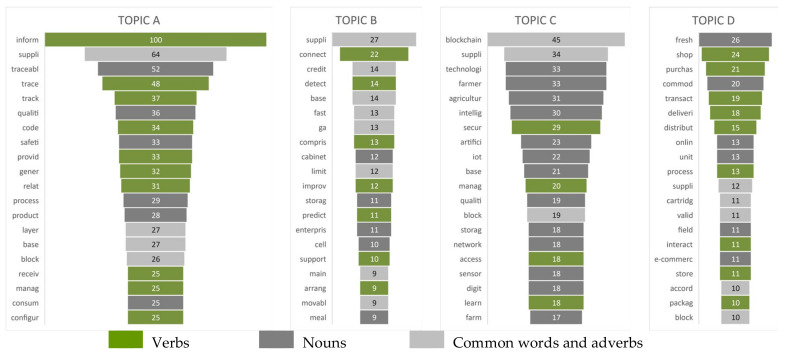
Frequencies of occurrence of the twenty most frequent keywords by topics.

**Figure 7 foods-12-01036-f007:**
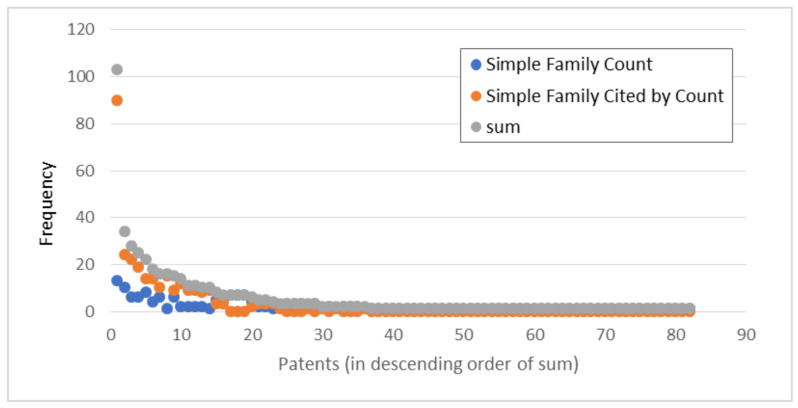
Assessment of the value of patents in the analyzed patent portfolio.

**Figure 8 foods-12-01036-f008:**
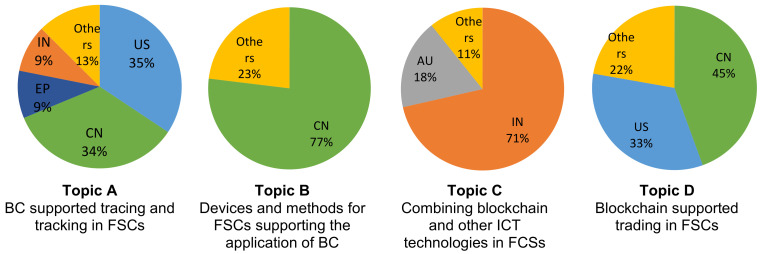
Distribution of patents within topics by country of origin.

**Table 1 foods-12-01036-t001:** Pseudocode of LDA generative process.

***choose*** a multinomial distribution *θd* for document *d (d ∈{1,…, M})* from a Dirichlet distribution with parameter *α* ***choose*** a multinomial distribution *βk* for topic *k (k ∈{1,…, K})* from a Dirichlet distribution with parameter *η* ***for*** each word *wn (n ∈{1,…, Nd })* in document *d* ***select*** a topic *Zn* from *θd* ***select*** a word *wn* from *βZk,n*

**Table 2 foods-12-01036-t002:** Input data for latent topic patent portfolio analysis and labeling.

Hidden Topic	Topic Label (Proposed by Authors)	Words among the 20 Most Common Words		Three Most Frequent IPCs	
		Verbs	Nouns	WIPO IPC (Frequency)	Description of IPC
**A**	**BC SUPPORTED TRACING AND TRACKING IN FSC**	inform trace track code provide generate relate receive manage configure	traceability quality safety process product consumer	G06Q10 (19)	Data processing systems: administration, management
				G06Q30 (11)	Data processing systems: commerce
				G06Q50 (7)	Systems or methods specially adapted for a specific business sector
**B**	**DEVICES AND METHODS SUPPORTING BC APPLICATION IN FSC**	connect detect comprise improve predict support arrange	cabinet storage enterprise cell meal	G06Q10 (5)	Data processing systems: administration, management
				G06F16 (5)	Digital computing or data processing equipment or methods, specially adapted for specific applications
				G06Q50 (4)	Systems or methods specially adapted for a specific business sector
**C**	**COMBINING BC AND OTHER ICT TECHNOLOGIES IN FSC**	secure manage access learn	technology artificial intelligence IoT base network sensor digit farmer agriculture quality storage farm	G06Q10 (25)	Data processing systems: administration, management
				G06Q50 (11)	Systems or methods specially adapted for a specific business sector
				G06Q30 (9)	Commerce, e.g. shopping or e-commerce
**D**	**BC SUPPORTED TRADING IN FSC**	shop purchase transact deliver distribute process interact store package	fresh commodity unit field online e-commerce	G06Q20 (9)	Payment architectures, schemes or protocols
				G06Q30 (8)	Commerce, e.g. shopping or e-commerce
				G06Q10 (5)	Data processing systems: administration, management

**Table 3 foods-12-01036-t003:** The 16 potentially most influential patents from the observed patent portfolio by identified topics.

Application Number	Title	Topic	Family count	Citation	Family + citation
Topic A: BC SUPPORTED TRACING AND TRACKING IN FSC					
EP2008875543	High-reliability product/activity tracking system	A	10	24	34
US16/386147	DNA Based Bar Code for Improved Food Traceability	A	6	19	25
US17/159123	Food Supply Tracking, Verification, and Feedback System	A	8	14	22
AU2020203178	A Machine Type Communication System or Device for Recording Supply Chain Information on a Distributed Ledger in a Peer to Peer Network	A	4	14	18
CN201811587331.4	Food tracing and query analysis system and method based on a HACCP system	A	1	15	16
EP2019892440	System, device, and process for tracking product	A	6	9	15
US17/257503	Enterprise Consumer Safety System	A	2	9	11
CN201610175359.1	Meat food supply chain tracing method based on RFID, QRCode and NFC	A	2	9	11
US15/378124	Supply chain tracking of farm produce and crops	A	2	8	10
US16/875011	Dot-matrix product information encoding for food traceability	A	5	3	8
**TOPIC B: DEVICES AND METHODS SUPPORTING APPLICATION OF BC IN FSC**					
CN201410151778.2	Internet fast food distribution cabinet	B	2	12	14
CN202010543066.0	Fresh food supply chain knowledge graph construction method based on semi-structured data	B	1	9	10
**TOPIC C: COMBINING BC AND OTHER ICT TECHNOLOGIES IN FSC**					
CA2776577	A system and method establishing an agricultural pedigree for at least one agricultural product	C	13	90	103
EP2019823567	Systems and methods for permission blockchain infrastructure with fine-grained access control and confidentiality-preserving publish/subscribe messaging	C	6	10	16
US16/978312	Non-specific, wireless detection of electrically or magnetically labeled bacteria and/or virus	C	4	3	7
**TOPIC D: BC SUPPORTED TRADING IN FSC**					
EP2017761847	System and method for determining or monitoring a process variable in an automation plant	D	6	22	28

## Data Availability

The data presented in this study are available on request from the corresponding author.

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
