# Peer review of "Emerging Perspectives of Blockchains in Food Supply Chain Traceability Based on Patent Analysis"

_foods, 2023, doi:10.3390/foods12051036_

Round 1

Reviewer 1 Report

In the paper, the authors analyze the prospects of blockchain technology (BC) for traceability in food supply chains (FSC). The authors use Patent Portfolio Analysis as a data source and apply Latent Dirichlet Allocation (LDA) modeling to identify the most relevant topics. From a base of 82 patents, the authors (i) identified trends in patent applications, showing that patent filing started in 2016 and since 2019 has shown exponential growth, with China, India, and the United States leading in the number of patents; (ii) modeled the technology portfolio having identified BC supporting tracing and tracking in FSCs, the development of devices and methods supporting the application of BC in FSCs, the combination of BC and other ICT Technology in FSCs, and BC-supported commercialization in FSCs as the four main topics under development; and (iii) provided a content analysis of the most relevant patents within each of the previous topics. The main results are discussed in light of the literature on the topic, and the main conclusions are presented.

The article deals with the relevant topic of traceability in food supply chains. Blockchain technology is fast developing, and its application is expanding rapidly from its original area (finance) to other areas, such as supply chains. One of the leading applications in supply chains has been traceability processes. The study brings interesting evidence by showing that the development of products and processes related to the application of BC in supply chain traceability has been growing considerably, with particular emphasis on the development of processes, devices, and methods combined with other ICTs. Therefore, the evidence presented reinforces the interest in BC and indicates the trends of innovations in this field and the main countries engaged in innovations. 

Overall, the manuscript is well-written and structured according to scientific standards. Thus, I suggest that the authors consider revising the points listed below to qualify the manuscript.

Title
Two points that I consider essential: (i) blockchain technology is central to the scope of the manuscript and is not explicit in the title, and (ii) Patent Portfolio Analysis (PPA) and Latent Dirichlet Allocation (LDA) have similar methodological relevance in the study, yet PPA is explicit in the title. I suggest the authors consider revising the title by including blockchain technology and reserving PPA and LDA for the abstract and method section.

Abstract
It would be opportune if the authors would advance in the abstract the main results of the study, especially on the identified four main topics.

Keywords
Almost all keywords are also present in the title or abstract. I suggest the authors consider revising the keywords using alternative terms that broaden the search and retrieval scope of the article.

Introduction
I liked the content of the introductory text. The authors managed to associate objectivity with the relevance of the topic.

Materials and Methods
a) Overall, the section is well written. However, I would recommend that the authors detail the steps or stages developed for LDA modeling for the sake of clarity for the reader. Although it is a relatively established methodology in the literature, not all readers have a sufficient grasp to understand the methodological process employed in the study. 

b) The content of lines 317 to 324 could be anticipated for this section.

Results
a) The complement 'patent analysis' is unnecessary for the section heading.
b) I suggest that the authors merge Figures 4 and 5 into a single figure (Figure 4, A, and B) to avoid an excessive number of figures in a relatively simple results topic. 
c) Lines 190 and 191: was the classification of verbs, nouns, common words, and adverbs done arbitrarily by the authors, or was it automated using specific software? The doubt reinforces my previous comment about the clarity of the method concerning LDA.
d) Figure 7: the space between the legend element and the corresponding label may confuse the interpretation of the figure.
e) The following excerpts from the text, Lines 235-242, 265-271, 286-293, and 307-311, added important information for understanding the content of the patents and what has been developed in FSC-applied CB. Very good!
f) Check the consistency of using bold text in identifying Topics A, B, C, and D.
g) Figure 8: it would be convenient if the authors added labels to the X and Y axes.
h) Figure 9: standardize the diameter of the figures.

Discussion

a) Adequate. The content contrasts results found with other findings reported in the scientific literature.

b) Line 430: "Authors agree...". Which authors? The authors of the manuscript or other authors in the scientific literature? Please clarify.

Conclusions

Adequate.

Although the text is well written, I recommend that the authors proofread the English once more.

Author Response

Dear reviewer,

First, we would like to thank you for your efforts to make our research more valuable. We are thankful also for the nice words regarding our work which supports our efforts to continue with additional motivation the research in this direction. On the other hand, your suggestion for improvements enabled us to improve a number of weak points in he first version of the manuscript. All changes made to the manuscript can be tracked via the Track Changes option.

We addressed your inputs in the following manner (new text provided in italic): 

Title

Remark 1.1:
Two points that I consider essential: (i) blockchain technology is central to the scope of the manuscript and is not explicit in the title, and (ii) Patent Portfolio Analysis (PPA) and Latent Dirichlet Allocation (LDA) have similar methodological relevance in the study, yet PPA is explicit in the title. I suggest the authors consider revising the title by including blockchain technology and reserving PPA and LDA for the abstract and method section.

Answer 1.1:

In order to improve manuscript title, and to emphasize blockchain original title “Emerging Perspectives of Food Supply Chain Traceability through Patent Portfolio Analysis” was changed to:

Emerging Perspectives of Blockchain in Food Supply Chain Traceability Based on Patent Analysis

Abstract

Remark 1.2:

It would be opportune if the authors would advance in the abstract the main results of the study, especially on the identified four main topics.

Answer 1.2:

Abstract was rewritten to include identified topics while remaining within word count limits:

In the field of blockchain (BC) technology application in the food supply chain (FSC), a patent portfolio is collected, described, and analyzed using Latent Dirichlet Allocation (LDA) modeling, with the aim of obtaining insight into technology trends in this emerging and promising field. A patent portfolio consisting of 82 documents was extracted from patent databases using PatSnap software. The analysis of latent topics using LDA indicates that inventions related to the applica-tion of BC in FSC are patented in four key areas: (A) BC-supported tracing and tracking in FSC, (B) devices and methods supporting application of BC in FSC, (C) combining BC and other ICT tech-nologies in FSC, and (D) BC-supported trading in FSC. Patenting of BC technology applications in FSC started during the second decade of the 21st century. Consequently, patent forward citation has been relatively low, while the family size confirms that application of BC in FSC is not yet widely accepted. A significant increase in the number of patent applications was registered after 2019, indicating that the number of potential users in FSC is expected to grow over time. The largest numbers of patents originate from China, India, and the US

Keywords

Remark 1.3:

Almost all keywords are also present in the title or abstract. I suggest the authors consider revising the keywords using alternative terms that broaden the search and retrieval scope of the article.

Answer 1.3:

Keywords were supplemented with the following words:

Food chain; traceability; blockchain application; database; patent portfolio; LDA, topic modeling

Remark 1.4:

Introduction
I liked the content of the introductory text. The authors managed to associate objectivity with the relevance of the topic.

Answer 1.4:

Thank you for this comment. We tried to be concise and explicit. However, based on the inputs form the other reviewer introduction was complemented in the following manner:

The first sentence in the introduction (lines 28-30) was changed in the following manner:

Food supply chain (FSC) comprising of all the stages through which food products are delivered to consumers, from the preparation of agricultural production, through storage, processing, distribution, and further towards food consumption is a complex system which includes different operators, resources and processes (Baralla, et al., 2019).

The following sentence were added in the introduction after lines 38:

Common to all supply chains is that they could benefit from implementing traceability [4], especially in reverse supply chains when unused, faulty, or damaged products are returned to manufacturer in an attempt to improve quality [5]. The weakness of all supply chains, including supply chains of pharmaceutical products, courier express, luxury goods, consumer electronics, manufactured goods, automobiles, textile, wood, dangerous goods, and particularly agri-food products is that transparent information is not currently offer to the final consumer as part of supply chain management [4].

The following sentence were added in the introduction (line 49):

Insight into the capabilities of blockchain-based traceability solutions come from the academic literature, in which it is pointed out that both tracking and tracing via BC ap-plications contribute to better transparency in supply chains and are thus gaining in popularity, particularly in food and pharmaceutical supply chains, the conclusion, however, is that in spite of numerous applications reported in the literature, most of them are conceptual in nature and real implementations of BC-based traceability solutions are very rare [4].

The following sentence were added in the introduction (line 57):

When proposing a framework for traceability within FSCs and addressing the key challenges for FSC stakeholders, Baralla et al. [1] emphasized BC in combination with IoT as technologies of choice for the development of such a system. An easy and simple methodology for integration of BC technology in FSCs, that allows traceability and provides the consumers with sufficient information to make informed purchasing decisions was proposed by Bettín-Díaz et al. [19] based on best practices engineering, technology, and marketing.

Materials and Methods

Remark 1.5:
a) Overall, the section is well written. However, I would recommend that the authors detail the steps or stages developed for LDA modeling for the sake of clarity for the reader. Although it is a relatively established methodology in the literature, not all readers have a sufficient grasp to understand the methodological process employed in the study. 

Answer 1.5:

More detailed  explanation regarding LDA was included by expansion of description in lines 110 to 113 I the following manner:

The pre-processed abstract texts were used to group patents in the portfolio into technology topics using Latent Dirichlet Allocation modeling (LDA) [45]. In LDA, the documents are viewed as random mixtures of different topics, where each topic is char-acterized according to the distribution of words. The topic distributions in documents share the common Dirichlet prior α, and the word distributions of topics share the common Dirichlet prior η. Given the parameters α and η for document d, parameter θd of a multinomial distribution over K topics is constructed from Dirichlet distribution θd|α. Similarly, for topic k, parameter βk of a multinomial distribution over all words is derived from Dirichlet distribution βk|η. The generative process of LDA method is given in Tab. 1.

Table 1. Pseudocode of LDA generative process

choose a multinomial distribution θd for document d (d {1,..., M})

            from a Dirichlet distribution with parameter α

choose a multinomial distribution βk for topic k (k {1,..., K})

            from a Dirichlet distribution with parameter η

for each word wn (n {1,..., Nd }) in document d

    select a topic Zn from θd

    select a word wn from βZk,n

Remark 1.6:
b) The content of lines 317 to 324 could be anticipated for this section.

Answer 1.6:

The text from lines 317 to 324 was moved to the section representing material and methods in the last paragraph (after line 144) which now states the following:

The outcome of the LDA was used to draw a dynamic picture of the development of BC technologies application in FSC, with analysis of geographical distribution, and identification of the most valuable patent applications. The value of patents was assessed based on its bibliographic attributes including forward citation and patent family size [32, 46-48]. In the cited works, influential patents having dominant technology or high applicability are distinguished and, accordingly, was used as an indicator of the patent’s technological significance. A simple patent family representing a set of patents applied in various countries to protect a single invention, besides representing the importance of technology, also reveals the width of its potential market [32, 46]. A larger patent family is indicative of the attractiveness of the protected invention or that the technology is more innovative. Obtained results were used as the base for identifying the potential drivers and paths in future development of BC applications in FSCs.

Results

Remark 1.7 (with answers):

  1. a) The complement 'patent analysis' is unnecessary for the section heading.

'patent analysis' was removed from section heading

  1. b) I suggest that the authors merge Figures 4 and 5 into a single figure (Figure 4, A, and B) to avoid an excessive number of figures in a relatively simple results topic.

Figures are now joined, and numbering for all subsequent figures is changed.  

  1. c) Lines 190 and 191: was the classification of verbs, nouns, common words, and adverbs done arbitrarily by the authors, or was it automated using specific software? The doubt reinforces my previous comment about the clarity of the method concerning LDA.

It is done manually arbitrarily by the authors in order to further easier understanding underlying trends. The explanation was added to the part describing Material and methods.

In line 135 the sentence added:

We manually classified the most frequent words, identified by LDA a nouns, verbs and adverbs and other words to enable understanding the common features for each topic.

  1. d) Figure 7: the space between the legend element and the corresponding label may confuse the interpretation of the figure.

Figure 7 is now improved, and the distances between color and label are now reduced

  1. e) The following excerpts from the text, Lines 235-242, 265-271, 286-293, and 307-311, added important information for understanding the content of the patents and what has been developed in FSC-applied CB. Very good!

Thank you!

  1. f) Check the consistency of using bold text in identifying Topics A, B, C, and D.

Consistency of BOLD text usage is now corrected.

  1. g) Figure 8: it would be convenient if the authors added labels to the X and Y axes.

Label are now added to both axes.

  1. h) Figure 9: standardize the diameter of the figures.

Figure is now improved and figures are now standardized.

Discussion

  1. a) Adequate. The content contrasts results found with other findings reported in the scientific literature.

Thank you!

Remark 1.8: 

  1. b) Line 430: "Authors agree...". Which authors? The authors of the manuscript or other authors in the scientific literature? Please clarify.

Answer 1.8:

In order to improve clarity, the fact that it is about authors in academic literature was emphasized in the sentence by moving citation to the point where authors are mentioned:

“AThe authors of [64] also  agree also that it is important that the introduction of blockchain technology in FSCs is not limited to this single technology, but, rather to, the whole stack of new technologies, including sensing technologies, big data analytics, arti-ficial intelligence, Internet of Things robotics, digital twins, cyber-physical systems, 3D printing and others”

Conclusions

Adequate.

Remark 1.9: 

Although the text is well written, I recommend that the authors proofread the English once more.

Answer 1.9:

The manuscript was proof-read by a native English speaker

Reviewer 2 Report

Overall this is an interesting research, however, several improvements are mandatory: 

1. The abstract needs to be restructured following the journal's guidelines.

2. The introduction section needs to be expanded, covering more related research in the field of traceability for the supply chain. Suggested papers to be referenced are:

Gayialis, S. P., Kechagias, E. P., Konstantakopoulos, G. D., & Papadopoulos, G. A. (2022). A Predictive Maintenance System for Reverse Supply Chain Operations. Logistics6(1), 4.

Sunny, J., Undralla, N., & Pillai, V. M. (2020). Supply chain transparency through blockchain-based traceability: An overview with demonstration. Computers & Industrial Engineering150, 106895.

Baralla, G., Pinna, A., & Corrias, G. (2019, May). Ensure traceability in European food supply chain by using a blockchain system. In 2019 IEEE/ACM 2nd International Workshop on Emerging Trends in Software Engineering for Blockchain (WETSEB) (pp. 40-47). IEEE.

Bettín-Díaz, R., Rojas, A. E., & Mejía-Moncayo, C. (2018). Methodological approach to the definition of a blockchain system for the food industry supply chain traceability. In Computational Science and Its Applications–ICCSA 2018: 18th International Conference, Melbourne, VIC, Australia, July 2-5, 2018, Proceedings, Part II 18 (pp. 19-33). Springer International Publishing.

3. The analysis is of good quality, however, it is only performed quantitatively. Additionally, the discussion section is too short and just summarizes the findings. The authors need to elaborate on the data and add more analysis by making valuable qualitative suggestions.

4. The authors don't explain the contribution of their research to the scientific community, nor do they discuss the limitations of their research.

5. The authors need to discuss similar research on this topic and explain how their research and results relate to them. 

6. In many cases, the authors write overly large sentences that are difficult to follow and contain multiple syntax and grammar errors that must be fixed.

Author Response

Dear reviewer,

First, we would like to thank you for your efforts to make our research more valuable. We are thankful also for the nice words regarding our work which supports our efforts to continue with additional motivation the research in this direction. On the other hand, your suggestion for improvements enabled us to improve a number of weak points in he first version of the manuscript. All changes made to the manuscript can be tracked via the Track Changes option.

We addressed recommended improvements in the following manner (new text provided in italic): 

Remark 2.1:

The abstract needs to be restructured following the journal's guidelines.

Answer 2.1:

The abstract is rewritten according to journals guidelines. Also, other reviewer requested incorporation of TOPCS, so whole ABSTRACT is rewritten and it now states the following:

In the field of blockchain (BC) technology application in the food supply chain (FSC), a patent portfolio is collected, described, and analyzed using Latent Dirichlet Allocation (LDA) modeling, with the aim of obtaining insight into technology trends in this emerging and promising field. A patent portfolio consisting of 82 documents was extracted from patent databases using PatSnap software. The analysis of latent topics using LDA indicates that inventions related to the applica-tion of BC in FSC are patented in four key areas: (A) BC-supported tracing and tracking in FSC, (B) devices and methods supporting application of BC in FSC, (C) combining BC and other ICT tech-nologies in FSC, and (D) BC-supported trading in FSC. Patenting of BC technology applications in FSC started during the second decade of the 21st century. Consequently, patent forward citation has been relatively low, while the family size confirms that application of BC in FSC is not yet widely accepted. A significant increase in the number of patent applications was registered after 2019, indicating that the number of potential users in FSC is expected to grow over time. The largest numbers of patents originate from China, India, and the US.

Remark 2.2:  

The introduction section needs to be expanded, covering more related research in the field of traceability for the supply chain. Suggested papers to be referenced are:

Answer 2.2:

We added the following text in the introduction:

The first sentence in the introduction (lines 28-30) was changed in the following manner:

The Ffood supply chain (FSC), comprising ofes all the stages through which food products are delivered to consumers, from the preparation of agricultural production, through storage, processing, distribution, and further towards food consumption, is a complex system which includes different operators, resources and processes [1].

The following sentence were added in the introduction after lines 38:

Common to all supply chains is that they could benefit from implementing traceability [4], especially in reverse supply chains when unused, faulty, or damaged products are returned to manufacturer in an attempt to improve quality [5]. The weakness of all supply chains, including supply chains of pharmaceutical products, courier express, luxury goods, consumer electronics, manufactured goods, automobiles, textile, wood, dangerous goods, and particularly agri-food products is that transparent information is not currently offer to the final consumer as part of supply chain management [4].

The following sentence were added in the introduction (line 49):

Insight into the capabilities of blockchain-based traceability solutions come from the academic literature, in which it is pointed out that both tracking and tracing via BC ap-plications contribute to better transparency in supply chains and are thus gaining in popularity, particularly in food and pharmaceutical supply chains, the conclusion, however, is that in spite of numerous applications reported in the literature, most of them are conceptual in nature and real implementations of BC-based traceability solutions are very rare [4].

The following sentence were added in the introduction (line 57):

When proposing a framework for traceability within FSCs and addressing the key challenges for FSC stakeholders, Baralla et al. [1] emphasized BC in combination with IoT as technologies of choice for the development of such a system. An easy and simple methodology for integration of BC technology in FSCs, that allows traceability and provides the consumers with sufficient information to make informed purchasing decisions was proposed by Bettín-Díaz et al. [19] based on best practices engineering, technology, and marketing.

Remark 2.3 and 2.4: 

  1. The analysis is of good quality, however, it is only performed quantitatively. Additionally, the discussion section is too short and just summarizes the findings. The authors need to elaborate on the data and add more analysis by making valuable qualitative suggestions.
  2. The authors need to discuss similar research on this topic and explain how their research and results relate to them. 

Answers to 2.3 and 2.4:

Both suggestions were addressed through extended discussion as follows:

BC technology is characterized by quite wide possibilities for application in different fields. Backman et al. [49] identified that beside financial transactions in cryptocurrencies, integrity verification in insurance, data management in human resources, privacy and security in secure storage, reputation of education, governance of e-voting, integrity of IoT business, electronic health records, and finally business industry supply chain are also prospective fields for BC application. For BC in FSCs as the subfield, the latest research by Zhao et al. [59] points out, based on findings from the academic literature, that applications have been developed in four main directions comprising, in addition to traceability, also manufacturing, water management and information security.

In the present research, we confirm that the most developed field of BC application in FSCs is related to traceability enhancement.

The following sentences were added to discussion (line 369):

An exponential increasing trend in BC-related patenting in general started in 2017 and reached 14072 patents in 2021 from 231 in 2016 [49]. Patent analysis in other areas shows similar trends but with difference in terms of patents targeting different fields of application. For example, BC on 5G patent portfolio consisted of over 1000 public and over 120 approved patents in the periods 2014 to 2020 [50], while for BC and energy, a study from 2020 revealed 24 approved and 295 public patents [51].

The following sentences were added to discussion (line 375):

The situation of BC technology application in other supply chains might be good indicator of probable development in FSCs. Patent life cycle analysis results indicate that patents related to BC based on 5G peaked in 2018, followed by a bottleneck period beginning in 2019 [50]. Patenting in the cluster of smart contract applications, privacy preserving and intellectual property, certificate issuance and verification, tokenization, virtual reality gaming and interoperability is also in a stage of maturity [49]. Such findings indicate that similar fast development might be expected also in the case of BC application in FSCs.

The following sentences were added to discussion (line 394):

The identified situation regarding geographical distribution of patents dealing with BC in FSCs is somewhat different from that of the general distribution of patents related to BC. In 2021, the largest share of BC-related patents originated from China (36 %), followed by US (26 %), while India represented only 2% of patents [49].

The following sentence were added in discussion after line 404:

The most frequently assigned IPC to the patents related to BC application in FSCs was G06Q for all identified latent topics. However, the analysis of 335 patents related to BC technology in general awarded in US as of 2020 [58] revealed that the most frequents IPC codes were H04L (transmission of digital information, e.g., telegraphic communica-tion) present in 84 % of patents, followed with G06F (electric digital data processing) present in 53 % of patents, while G06Q (data processing systems or methods, specially adapted for administrative, commercial, financial, managerial, supervisory or forecasting purposes) was assigned to 33 % of analyzed patents. This observation points out the somewhat specific focus of patents related to BC in FSCs.

The following sentences were added to discussion (line 469):

This observation is supported also through the fact that almost half of the initiatives and projects registered in the field of BC application in FSC [70] is targeting food integrity.

The following sentences were added in discussion after line 475:

The fact that BC technology is attracting the attention of numerous stakeholders is also confirmed through the existence of many projects and initiatives that aim to con-tribute to the establishment of a trusted environment, are the building of more sustainable, more transparent, and more integrated FSCs. Based on the analysis of projects and initiatives related to BC in FSCs targeting FSC issues, besides the dominant issue of food integrity (50 % of identified initiatives), there were also issues of supervision and man-agement of FSCs (14.5 % of identified initiatives), waste reduction and environmental awareness (10 %), support of small farmers (16 %), food safety (6 %) and food security (4 %) [71].

The following sentences were added in discussion, line 483:

However, the development of BC applications in FSCs is closely related to integration with other information and communication technologies, in particular IoT. Finding that combination of BC technology with other information and communication technology and IoT in particular has been adopted for the improvement of in FSC, was already emphasized in [59].

Remark 2.5:

The authors don't explain the contribution of their research to the scientific community, nor do they discuss the limitations of their research.

Answer 2.5:

The following paragraphs were added at the end of discussion, after line 483:

. A most recent study providing content-based analysis of the academic literature [77] revealed that BC application in the agrifood sector is a relatively recent topic that started to gain traction in 2018, when the research community started intensively dealing with this issue, with more than 75 % of identified related records in the academic literature being published in 2020 and 2021. The identified sharp increase in the number of patents and patent applications related to BC application in FSCs is a reflection of vigorous research activities resulting in ready-to-implement solutions in this field. The obtained results will support research community members in making decisions related to the development of high readiness level products in the field of BC application in FSCs.

The main limitation related to present research is the lag period of patents which has certainly contributed to incomplete results within the past year. The second limitation is related to the fact that patent application, does not necessarily result in approved patent were taken into consideration, but due to low number of approved patents they had to be taken into account for this analysis. However, the clearly identified increasing trend in the number of patents suggests that these limitations most probably do not affect the obtained results to a great extent.

As the result of introduction and discussion extension new referencea were included. Reference part was supplemented with additional reference used in the discussion and introduction part as follows:

  1. Gayialis, S. P., Kechagias, E. P., Konstantakopoulos, G. D., & Papadopoulos, G. A. (2022). A Predictive Maintenance System for Reverse Supply Chain Operations. Logistics6(1), 4.
  2. Sunny, J., Undralla, N., & Pillai, V. M. (2020). Supply chain transparency through blockchain-based traceability: An overview with demonstration. Computers & Industrial Engineering150, 106895.
  3. Baralla, G., Pinna, A., & Corrias, G. (2019, May). Ensure traceability in European food supply chain by using a blockchain system. In 2019 IEEE/ACM 2nd International Workshop on Emerging Trends in Software Engineering for Blockchain (WETSEB)(pp. 40-47). IEEE.
  4. Bettín-Díaz, R., Rojas, A. E., & Mejía-Moncayo, C. (2018). Methodological approach to the definition of a blockchain system for the food industry supply chain traceability. In Computational Science and Its Applications–ICCSA 2018: 18th International Conference, Melbourne, VIC, Australia, July 2-5, 2018, Proceedings, Part II 18(pp. 19-33). Springer International Publishing.
  5. Gao, F., Chen, D. L., Weng, M. H., & Yang, R. Y. (2021). Revealing development trends in blockchain-based 5g network technologies through patent analysis. Sustainability, 13(5), 2548.
  6. Huang, L. Y., Cai, J. F., Lee, T. C., & Weng, M. H. (2020). A study on the development trends of the energy system with blockchain technology using patent analysis. Sustainability, 12(5), 2005.
  7. Bamakan, S. M. H., Bondarti, A. B., Bondarti, P. B., & Qu, Q. (2021). Blockchain technology forecasting by patent analytics and text mining. Blockchain: Research and Applications, 2(2), 100019.
  8. Dal Mas, F., Massaro, M., Ndou, V., & Raguseo, E. (2023). Blockchain technologies for sustainability in the agrifood sector: A literature review of academic research and business perspectives. Technological Forecasting and Social Change, 187, 122155.
  9. Zhao, G., Liu, S., Lopez, C., Lu, H., Elgueta, S., Chen, H., & Boshkoska, B. M. (2019). Blockchain technology in agri-food value chain management: A synthesis of applications, challenges and future research directions. Computers in industry, 109, 83-99.
  10. Kamilaris, A., Fonts, A., & Prenafeta-Boldύ, F. X. (2019). The rise of blockchain technology in agriculture and food supply chains. Trends in Food Science & Technology, 91, 640-652.
  11. Yang, Y. J., & Hwang, J. C. (2020). Recent development trend of blockchain technologies: A patent analysis. International Journal of Electronic Commerce Studies, 11(1), 1-12.

Remark 2.6:

In many cases, the authors write overly large sentences that are difficult to follow and contain multiple syntax and grammar errors that must be fixed.

Answer 2.6:

Text was proof-read by a native English speaker

Round 2

Reviewer 1 Report

Dear Authors,

I have gone through all the responses and improvements you indicated in the response letter and highlighted them in the revised version of the manuscript.
I have replied to each of your responses in blue text.
There remain only two or three minor points that you might consider.
For the rest, I congratulate the authors on the improvements in the manuscript.
Please take a look at the attached document.

Author Response

Dear reviewer,

First of all, we want to express my sincere gratitude for your patience and diligence during the correction of our manuscript.

We deeply appreciate your willingness to work with us through a second round of corrections, and we apologize for any oversights on our part during the initial review process.

We are also thankful for Your dedication to improve our work, since it is now much more scientifically sound and it will, we are certain, make much more impact.

We separate answers that are not corrected properly:

keywords

„Reviewer's reply: Right! However, many keywords remain duplicates in the title and abstract. It is not a problem per se, except that it restricts the search scope. ‘Food safety’ could be an interesting keyword, for instance“

We added “food safety” and also “food authenticity” to the list of keywords.

1.7 In line 135 the sentence added:

„We manually classified the most frequent words, identified by LDA a nouns, verbs and adverbs and other words to enable understanding the common features for each topic.

Reviewer's reply:

“ Okay, but how could I find the same results by applying the same method to the same patent portfolio? Just a question to think about the limits to the replicability of the study.

Response: Our opinion is that this part of work is perfectly replicable, since Standard English division on word classes were used. Namely, LDA for the same portfolio will result in the same list of the most frequent words, and identification if they are nouns, verbs or adverbs is the matter of usual word classification which is applied in standard English. Anyway, word classification was only an auxiliary tool which contributed to easier and faster understanding of hidden content of identified topic. The same result could be reached also if the words were not divided to mentioned groups. However, we prefer to keep the classification in order to present to other authors involved in similar analysis the access to this simple approach which, according to our experience, contributed much to easier understanding of hidden topics content.

Figure 8: it would be convenient if the authors added labels to the X and Y axes.

Label are now added to both axes.

Reviewer's reply: Not good enough for me! The labels for Y-axis should be the legend. What I understand is the label for Y-axis should be only “Frequency”. Am I right?

You are absolutely right! We are very sorry for the oversight. Figure 8 is now corrected according to your suggestions.

Answer 1.8:

In order to improve clarity, the fact that it is about authors in academic literature was emphasized in the sentence by moving citation to the point where authors are mentioned: “The authors of [64] also agree also that it is important that the introduction of blockchain technology in FSCs is not limited to this single technology, but, rather to, the whole stack of new technologies, including sensing technologies, big data analytics, artificial intelligence, Internet of Things robotics, digital twins, cyber-physical systems, 3D printing and others”

Reviewer's reply: Why not “Renda [64] agrees also that it is important ….”? According to the respective reference (64) in the references list, there is only one author (Renda). So, “authors agree” is incorrect.

We appreciated your comment. Suggested correction will improve clarity. Sentence was corrected to:

Renda [64] agrees that benefits of the introducing BC technology in FSCs may address much wider issues related to food security and safety and human health in general, reduction of food waste and losses, animal welfare and climate change [65], as well as the development and digitalization of rural areas [66].

Reviewer 2 Report

The authors have greatly improved their paper

Author Response

Dear reviewer,

We are writing you to express gratitude for your review of our manuscript. Your insightful feedback and valuable suggestions have significantly improved the quality of our work. Your contributions have helped us to refine our text and make it more impactful.

We appreciate the time and effort you dedicated to carefully reviewing our manuscript. Your expertise and attention to detail have been incredibly valuable to us.